# Repetitive in vivo manual loading of the spine elicits cellular responses in porcine annuli fibrosi

John Robert Matyas[1]*, Claudia Klein[2¤], Dragana Ponjevic[1], Neil A. Duncan[3], Gregory N. Kawchuk[4]

1 Department of Comparative Biology & Experimental Medicine, McCaig Institute of Bone and Joint Health, University of Calgary Faculty of Veterinary Medicine, Calgary, Alberta, Canada, 2 Department of Clinical and Veterinary Clinical Sciences, Faculty of Veterinary Medicine, University of Calgary, Calgary, Alberta, Canada, 3 Department of Civil Engineering, Schulich School of Engineering, University of Calgary, Calgary, Alberta, Canada, 4 Department of Physical Therapy, Faculty of Rehabilitation Medicine, University of Alberta, Edmonton, Alberta, Canada

¤ Current address: Institute of Farm Animal Genetics, Friedrich-Loeffler-Institut, Federal Research Institute for Animal Health, Neustadt, Germany

* jmatyas@ucalgary.ca

**Data Availability Statement:** The complete list of up- and down-regulated genes are available in Supplementary S1 and S2 Tables. Sequencing data are available on the NCBI Gene Expression

## Abstract

Back pain and intervertebral disc degeneration are prevalent, costly, and widely treated by manual therapies, yet the underlying causes of these diseases are indeterminate as are the scientific bases for such treatments. The present studies characterize the effects of repetitive in vivo manual loads on porcine intervertebral disc cell metabolism using RNA deep sequencing. A single session of repetitive manual loading applied to the lumbar spine induced both up- and down-regulation of a variety of genes transcribed by cells in the ventral annuli fibrosi. The effect of manual therapy at the level of loading was greater than at a level distant to the applied load. Gene ontology and molecular pathway analyses categorized biological, molecular, and cellular functions influenced by repetitive manual loading, with over-representation of membrane, transmembrane, and pericellular activities. Weighted Gene Co-expression Network Analysis discerned enrichment in genes in pathways of inflammation and skeletogenesis. The present studies support previous findings of intervertebral disc cell mechanotransduction, and are the first to report comprehensively on the repertoire of gene targets influenced by mechanical loads associated with manual therapy interventions. The present study defines the cellular response of repeated, low-amplitude loads on normal healthy annuli fibrosi and lays the foundation for future work defining how healthy and diseased intervertebral discs respond to single or low-frequency manual loads typical of those applied clinically.

## Introduction

Persistent low back pain is a global problem responsible for more years-lived-with-disability than any other condition [1, 2]. Hence, the health, societal, and economic burdens associated

Omnibus repository https://www.ncbi.nlm.nih.gov/geo/, (Accession no. GSE166656). The original data for PCR experiments, DAVID, PANTHER, and WGCNA analyses are available at the Open Science Framework (OSF) data repository (DOI: 10.17605/OSF.IO/75NUG).

**Funding:** These studies were supported by a NIH grant 5R21AT4055-2 to authors GK and NAD. https://www.nih.gov/ Funders did not play a role in study design, data collection, decision to publish, or manuscript preparation.

**Competing interests:** The authors have declared that no competing interests exist.

with persistent low back pain approximate those of cardiovascular disease, cancer, mental illness, and autoimmune diseases [3]. However, the causes of persistent back pain are poorly understood and, as a consequence, many different treatments are recommended to patients, including surgical and conservative interventions. Given that many common treatments also have the potential to produce significant harm (e.g., chronic NSAIDS, opioids, surgery), conservative interventions for back pain are worthy of full consideration. Delivered by various health professionals, including physical therapists, athletic therapists, osteopaths, and most commonly chiropractors, spinal manipulative therapy is a conservative intervention that is recommended by several guidelines for people with chronic back pain [4–6]. The procedure entails rapidly applying manual force to the external surface of the back with the intention of improving spinal musculoskeletal function and reducing pain. Previous experimental studies on pig spines, using carefully calibrated robotic movements to reproduce clinical manual therapy, indicate that when manual force is applied to the spine, the intervertebral discs (IVD) receive the largest fraction of the applied load of all spinal tissues [7], and manual loading applied to the dorsal spine is expected to induce maximal loading in the ventral annuli fibrosi. Nevertheless, it remains unclear how applying forces for such a short duration might trigger either immediate or sustained biological responses in IVD tissues. Qualifying and quantifying biological responses of any kind after spinal manual therapy is an important first step in understanding how manual force might transduce a local therapeutic effect in the spine.

It is noteworthy that, in a number of clinical and pre-clinical studies, external mechanical loads reportedly transduce biological changes in the IVD. For example, clinical surgery designed to correct excessive spinal curvature (scoliosis) imparts sustained, large-magnitude forces to the IVD that induces tissue remodeling [8], which has been confirmed experimentally in animals [9]. Similarly, experimental implantation of special loading devices that apply static and dynamic, compressive or distractive, forces across spinal segments of rats and mice have been shown to induce demonstrable biological changes in the IVDs [9–12]. Moreover, mechanical loading of IVDs in organ culture induces alterations in matrix metabolism [13–15] as it does in IVD cells isolated, cultured, and loaded in vitro [16]. In all these experiments, load elicits biochemical, biophysical, or biological responses in the IVD that may, in turn, influence other systems. For example, it has been reported recently that spinal manipulation reduces spinal stiffness, changes disc diffusion, alters muscle function [6] and may possibly induce changes in neurophysiology (e.g., see review [17]). In these studies the possible role of IVD nerve endings is unknown, yet the presence of nerves in the outer annulus of normal [18] and degenerating discs [19] plausibly implicates the annuli fibrosi in spinal nociception.

Whereas a number of physiological processes may alter the biochemistry and mechanical function of the extracellular matrix in the long-term (e.g., collagen cross-linking or glycation), the primary short-term biological response of the IVD to mechanical loading is most likely to occur in the cells. Mechanotransduction, the transduction of an external mechanical stimulus into cellular activity, can be initiated by transcription of nucleic acids encoding for downstream translation of proteins that can serve as structural (e.g., collagens and proteoglycans) or regulatory (e.g., enzymes) roles in extracellular matrix biosynthesis and assembly, as cytokines involved in immune regulation and inflammation, as stimuli for membrane depolarization, and as promoters of gene activation or inhibition. Indeed, a remarkable mechanobiology study reports responses by intervertebral disc cells to such IVD loads detectable from even a *single* (1.5 hr) loading event [20], including both upstream (i.e., altered gene expression) as well as downstream effects (i.e., altered tissue concentrations of extracellular matrix proteoglycan and collagen). Notably, these authors report that mRNA changes in annuli fibrosi persist, whereas those in nucleus pulposus were transient [20]. The present experiments seek to explore the sensitivity of ventral annuli fibrosi cells—the cells responsible for maintaining the

structural integrity of the disc—in responding to loads typical of those in clinical practice when applied repeatedly by in vivo spinal manipulation of a porcine model using a discovery-based, broad-spectrum gene expression analysis.

Before it is possible to credibly interpret a "clinical dose" of manual therapy (i.e., the episodic application of short-term loads), it is necessary to define positive and negative controls to frame the interpretation of clinical loading paradigms. The present study seeks to identify any gene candidates of manual therapy-induced mechanotransduction by evaluating differences between the gene expression of IVD cells in positive and negative controls. In this initial study, a porcine model is used to define any changes in gene expression, as indices of cell responsiveness, in healthy normal annuli fibrosi of positive controls (treated with multiple applications of manual therapy over several hours) and sham-treated (non-loaded) negative controls. These initial proof-of-principle studies intend to serve as a foundation for future gene expression analyses of normal and diseased annuli fibrosi loaded using clinical doses of manual therapy of the spine.

## Results

The mean (±SD) yield of total RNA was 0.089 ± 0.035 (μg/mg wet weight) for the central ventral wedge of the $L_{3-4}$ annuli fibrosi. There was no significant difference in RNA yield among intervertebral discs between Hyperloaded and Control specimens (p = 0.49).

### RNA deep sequencing

After quality control (see Methods), 17.7 million raw reads per sample were aligned on average, revealing 13,402 unique expressed transcripts; of these, 11,370 were annotated sufficiently for DAVID analysis. Following filtering for transcripts, unsupervised hierarchical cluster analysis (JMP Genomics) detected two distinct groups that corresponded to Non-loaded Controls and Hyperloaded groups as evidenced clearly on the heatmap of raw data, which is presented with a sorted dendrogram of transcripts of similar expression profiles (Fig 1). Using a cutoff of p<0.01, annuli fibrosi tissues treated with Hyperloading, compared to Non-loaded Controls, were calculated to have 348 mRNAs significantly up-regulated, 430 mRNAs significantly down-regulated, with 10590 mRNAs identified as "background," (i.e., transcribed mRNAs with levels p≥0.01). A Principal Component Plot reveals close clustering of Hyperloaded samples implying that any effects of loading duration or frequency are indistinguishable (Fig 2).

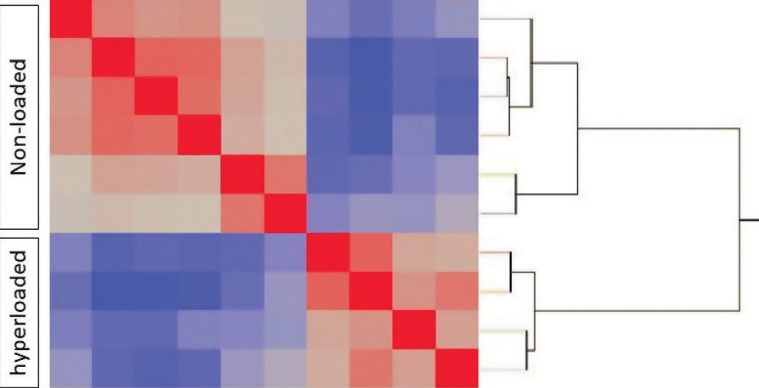

**Fig 1. Heat map of up- and down-regulated transcripts mapped with a dendrogram (red = up-regulated [n = 348]; blue = down-regulated [n = 430]; grey = unchanged [n = 10590]).** The changes in gene expression self-segregated to form distinct quadrants for Non-loaded and Hyperloaded groups.

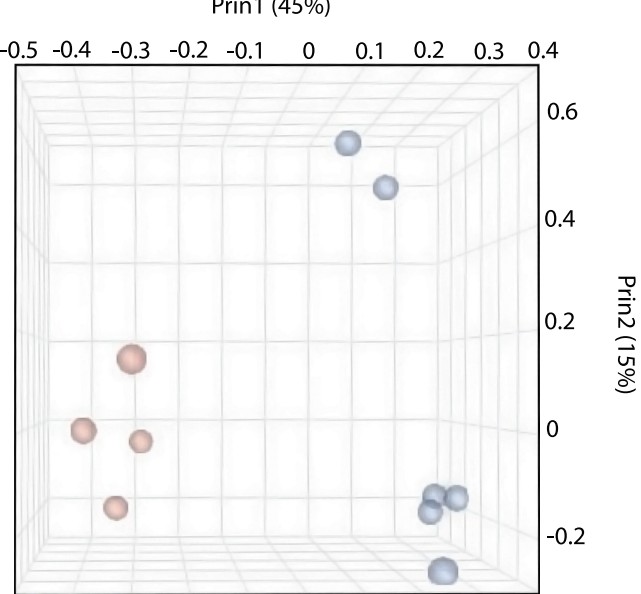

**Fig 2. Principal component analysis (PCA) depicts some variation amongst Non-loaded controls (blue) possibly related to animal age or body mass, yet reveals a noteworthy and distinct segregation of Non-loaded controls from Hyperloaded (red) samples.** Close grouping of Hyperloaded samples infers negligible effects of loading duration and frequency.

Table 1 lists, by fold-change in transcript expression, an assortment of 20 differentially expressed transcripts selected by magnitude of change and relevance to IVD pathobiology. The full lists of up- and down-regulated genes are provided online as S1 and S2 Tables.

### RT-qPCR analysis of select RNA sequencing targets

RT-qPCR results are listed as fold-change in mRNA copy number in Non-loaded Control and Hyperloaded samples and are listed in comparison to RNA sequencing fold-change (Table 2). Note the relatively diminished response to Hyperloading in the $L_{1-2}$ versus $L_{3-4}$ discs.

### Functional analyses of RNAseq transcripts

DAVID Functional Annotation analysis enumerated transcript count and the percentage of differentially expressed transcripts, of which the top 20 (of 237) are listed in Table 3 along with their calculated Enrichment scores.

PANTHER Functional Analyses plot the percentage of listed background, up-, and down-Regulated genes for various Biological Processes, Molecular Functions, and Cellular Components (Fig 3) using Fisher's Exact test with the FDR multiple-test correction (n = 348 up-regulated; n = 430 down-regulated, and n = 10590 unchanged genes). The numbers of up-regulated and down-regulated genes were highly correlated for each of these analyses (>0.95) and both were highly correlated (>0.92) to the number of background genes (i.e., the number of transcribed genes in each category). Compared to background expression, the *relative* over- and under-representation of differentially expressed transcripts in Gene Ontogeny (GO) classes of various biological and cellular processes is given in Table 4.

WGCNA (Weighted Gene Co-expression Network Analysis) of differentially expressed transcripts and clinical traits Treatment, Sex, and Body Mass, revealed two major modules of gene networks (blue [240 genes] and turquoise [280 genes] mapped with dendrogram—Fig 4)

**Table 1. Twenty selected transcripts with differential expression possibly relevant to IVD pathobiology.**

| Gene | Gene description | Fold-Change* | p-Value | Function (https://www-ncbi-nlm-nih-gov/gene) |
|------|-----------------|--------------|---------|----------------------------------------------|
| CCL8 | chemokine ligand 8 | 49.46 | 0.0026 | Immunoregulatory/Inflammatory |
| CCL2 | C-C motif chemokine 2 | 26.34 | 0.0018 | Immunoregulatory/Inflammatory |
| PTPRO | protein tyrosine phosphatase, receptor type O | 16.96 | 0.0035 | Polarized cell membrane. . . |
| CCL24 | C-C motif chemokine ligand 24 | 14.42 | 0.0045 | Immunoregulatory/Inflammatory |
| TNMD | tenomodulin | 12.91 | 0.0073 | Angiogenesis inhibitor |
| ADAMTS4 | ADAM metallopeptidase with thrombospondin type 1 motif 4 | 10.64 | 0.0115 | Aggrecanase |
| CCL4 | C-C motif chemokine 4 | 8.49 | 0.0002 | Immunoregulatory/Inflammatory |
| SRGN | serglycin | 6.15 | 0.0025 | Hematopoietic secreted proteoglycan/apoptosis |
| IGF1 | insulin like growth factor 1 | 5.11 | 0.0110 | Growth and development |
| ADAMTS1-201 | A disintegrin and metalloproteinase with THBS motifs 1 precursor | 4.85 | 0.0124 | Inflammation and angiogenesis inhibitor |
| GALR3 | galanin receptor 3 | -3.85 | 0.0007 | Receptor for galanin involved in cognition and pain |
| ACAN | aggrecan | -5.26 | 0.0088 | Large aggregating proteoglycan of cartilage |
| COL11A2 | collagen type XI alpha 2 chain | -5.56 | 0.0064 | Structural co-polymer with Col2 in cartilage |
| KCNA1 | potassium voltage-gated channel subfamily A member 1 | -10.00 | 0.0014 | Voltage-gated potassium channel |
| FBXO2 | F-box protein 2 | -10.00 | 0.0034 | Phosphorylation-dependent ubiquitination |
| Slc12a2 | solute carrier family 6 member 12 | -11.11 | 0.0039 | Ion balance; cell volume regulation |
| TPD52L1 | tumor protein D52-like 1 | -12.50 | 0.0087 | Cell proliferation, calcium signalling, apoptosis |
| GAL3ST1 | galactose-3-O-sulfotransferase 3 | -14.29 | 0.0000 | Membrane glycolipid sulfation/myelin |
| EPS8L2 | EPS8 like 2 | -14.29 | 0.0069 | Growth factor driven actin re-organization |
| CILP | cartilage intermediate layer protein | -14.29 | 0.0051 | IGF-1 antagonist |

* Fold-change positive = up-regulated; negative = down-regulated in Hyperloaded versus Non-loaded Controls.

that strongly associated with the trait Treatment (Non-loading, Hyperloading—Fig 5). Based on calculated enrichments for Gene Ontology, the top 10 ranked transcripts for each module are listed in Table 5. The blue module is enriched with genes that participate principally in inflammatory processes; the turquoise module is enriched with genes that participate principally in skeletogenesis.

## Discussion

Spinal diseases, particularly low back pain, are commonly treated with some form of physical intervention. Even if the underlying mechanisms of these physical interventions are unknown and potentially complex, clinical reports suggest such interventions may influence low back pain in some, though not all, people [21]. Hence, it is unsurprising that extensive clinical and basic research has investigated the influence of physical, i.e., mechanical, loads on the spine

**Table 2. Normalized fold-change in Hyperloaded discs of select RT-PCR targets.**

| Group (Normalized to Control = 1) | CCL8 | CCL2 | AQP9 | SRGN | CILP | COL11A | FBLN7 |
|-----------------------------------|------|------|------|------|------|--------|-------|
| Control (n = 6) | 1.00 | 1.00 | 1.00 | 1.00 | 1.00 | 1.00 | 1.00 |
| L3,4 Hyperloaded (n = 4) | 8.12 | 4.09 | 4.24 | 4.23 | 0.78 | 1.33 | 0.75 |
| L1,2 Hyperloaded (n = 3)* | 2.01 | 2.20 | 0.96 | 1.56 | 0.55 | 1.25 | 0.68 |
| RNAseq Fold-Change (L3,4)** | 40.06 | 26.74 | 3.18 | 5.45 | 0.06 | 0.36 | 0.46 |

*Paired sample t-test for seven RT-PCR targets reveals higher treatment effect in ventral annuli samples from $L_{3,4}$ (directly under loading site) compared to $L_{1,2}$ (two segments proximal to loading site) (n = 3) p = 0.043

**Correlation coefficient between fold-change of RT-PCR and RNAseq for seven targets in $L_{3,4}$ ventral annuli (n = 10) $R^2$ = 0.75

**Table 3. DAVID: Top 20 clusters of differentially expressed RNA transcripts.**

| Annotation Cluster | Enrichment Score | Cellular Functions |
|---|---|---|
| Annotation Cluster 1 | 5.56 | Membrane/transmembrane |
| Annotation Cluster 2 | 5.35 | Chemotaxis/cytokine |
| Annotation Cluster 3 | 3.87 | Immune Chemotaxis |
| Annotation Cluster 4 | 2.84 | Rhodopsin; G-coupled receptor |
| Annotation Cluster 5 | 1.83 | Leucine-rich repeat |
| Annotation Cluster 6 | 1.81 | Toll-Like Receptor |
| Annotation Cluster 7 | 1.61 | Collagen/complement |
| Annotation Cluster 8 | 1.30 | Metalloproteinase/peptidases |
| Annotation Cluster 9 | 1.28 | Sushi |
| Annotation Cluster 10 | 1.14 | G-protein signalling |
| Annotation Cluster 11 | 1.06 | Transplantation antigens |
| Annotation Cluster 12 | 0.99 | FERM |
| Annotation Cluster 13 | 0.89 | Protein kinase inhibition |
| Annotation Cluster 14 | 0.82 | Protein tyrosine phosphatase |
| Annotation Cluster 15 | 0.77 | Rho GTPase |
| Annotation Cluster 16 | 0.60 | Transmembrane |
| Annotation Cluster 17 | 0.53 | Zinc-finger |
| Annotation Cluster 18 | 0.53 | Cardiomyopathy |
| Annotation Cluster 19 | 0.25 | Protein kinase |
| Annotation Cluster 20 | 0.10 | SH3 |

pathophysiology. While pathology of the spine, including the intervertebral disc, is commonly documented in patients with a history of back pain, the exact source of back pain remains indeterminate, and it is unclear if manual therapies have any influence on IVD metabolism that might promote repair, preserve health, or diminish pain. This initial study defines changes from baseline gene expression in healthy discs in response to manual loads of magnitudes relevant to clinical treatment.

As with all experiments, these studies have several assumptions and limitations as well as certain advantages. Although the use of skeletally immature, quadrupedal pigs as a model of human IVD biology is an obvious limitation, the size of the pigs in this study enables both precise sampling of IVD tissues and the relevant application of spinal manipulation by an experienced clinician, which makes this model highly advantageous for the purposes of this study. The goal here was to define a positive control for applying compressive manual loading to normal healthy IVD, which leaves future studies to determine, whether or not, any molecular changes might occur after a typical, clinically applied spinal manipulation in healthy and diseased IVDs. Nevertheless, the mRNA changes documented here indicate that IVD cells have a distinct, near-to-short-term (within 4 hours) stimulation of gene transcription in response to applied, repetitive loads, which is in accord with the studies of MacLean et al. [20] and others. It must also be acknowledged that although an equal magnitude of load was applied to all Hyperloaded pigs, two subgroups received different frequencies of load, which would be an insufficient number to discern any graded effect of load. Nevertheless, in the present study, all pigs were entered as individuals into an unsupervised analysis, which statistically determined that they belonged in two distinct groups that corresponded to the binary variable of load, i.e., hyperloaded versus non-loaded individuals. Principal component analysis supports a distinct treatment effect of loading, but no appreciable differences of load history. Lastly, it is noteworthy that the present report pertains only the ventral segment of annuli fibrosi of healthy spines,

## PANTHER GO-Slim Functional Analyses

**Fig 3. PANTHER GO-Slim results for up-regulated and down-regulated genes analyses for functional networks of biological processes, molecular function, and cellular component.** Note the relative over-representation of regulatory genes (membrane, catalysis, response, regulation) compared to structural genes (extracellular matrix, structural molecule, adhesion) immediately after Hyperloading.

and based on previously published work in humans [22, 23] and bovines [24], it seems likely that different parts of the intervertebral disc might receive different types and magnitudes of loads and would likely have different cellular responses based on region and health of the disc.

While it is well known that the skeleton responds reliably to repeated loads with biological adaptations (e.g., muscle tone and skeletal density) over long-time scales (weeks-to-months), short-term loads activate cellular metabolism as evidenced by changes in nucleic acid transcription [20], which can be detected and evaluated very precisely by RNA sequencing and RT-qPCR. In the present study, it is noteworthy that the yield of RNA extracted from annuli fibrosi is similar to previous reports of other dense connective tissues such as ligament and tendon [25]. And, of the reverse-transcribed mRNA, 6.8% (778 of 11,368 total transcripts) of ventral annulus transcripts were determined (by unsupervised clustering) to be differentially expressed between Non-loaded Controls and Hyperloaded groups. That transcripts were both up-regulated *and* down-regulated suggests that a systematic bias is unlikely to account for such differential expression. Moreover, the confirmation by RT-qPCR of transcript changes that generally mimic those of RNA sequencing (Table 2, $R^2 = 0.75$), and a "dose-responsive" effect of loading ($L_{3-4} > L_{1-2}$; $p < 0.05$) supports a genuine biological response to manual loading. The discrete quadrants readily visible on the RNA sequencing heatmap (Fig 1) and the segregation on the PCA plot (Fig 2) highlight the distinct differential expression of RNA

**Table 4. Over- and under-representation of PANTHER Gene Ontology (GO) activities compared to background.**

| Gene Ontology Activity | GO number | Representation compared to Background |
|---|---|---|
| **GO Biological Process** | | |
| RNA metabolic | GO:0016070 | Under- |
| Nucleobase-containing Compound Metabolic | GO:0006139 | Under- |
| Primary Metabolic | GO:0044238 | Under- |
| Nervous System Development | GO:0007399 | Over- |
| Single-multicellular Organism | GO:0044707 | Over- |
| System Development | GO:0048731 | Over- |
| **GO Molecular Function** | NA | NA |
| **GO Cellular Component** | | |
| Cell part | GO:0044464 | Under- |
| Cytoplasm | GO:0005737 | Under- |
| Intracellular | GO:0005622 | Under- |
| Organelle | GO:0043226 | Under- |
| Ribonucleoprotein Complex | GO:0030529 | Under- |
| Extracellular Region | GO:0005576 | Over- |
| Plasma Membrane | GO:0005886 | Over- |

transcripts between the Hyperloaded and Non-loaded Control groups, supporting an unbiased and unexpectedly clear treatment effect of a single session of repeated manual loading.

The selection of up-regulated transcripts in Table 1 reveal a preponderance of chemokines and enzymes with catabolic actions on extracellular matrix, while the down-regulated transcripts are notable for diminished expression of the (anabolic) structural (extracellular) matrix components aggrecan and collagen XI. If persistent, such changes infer net overall matrix

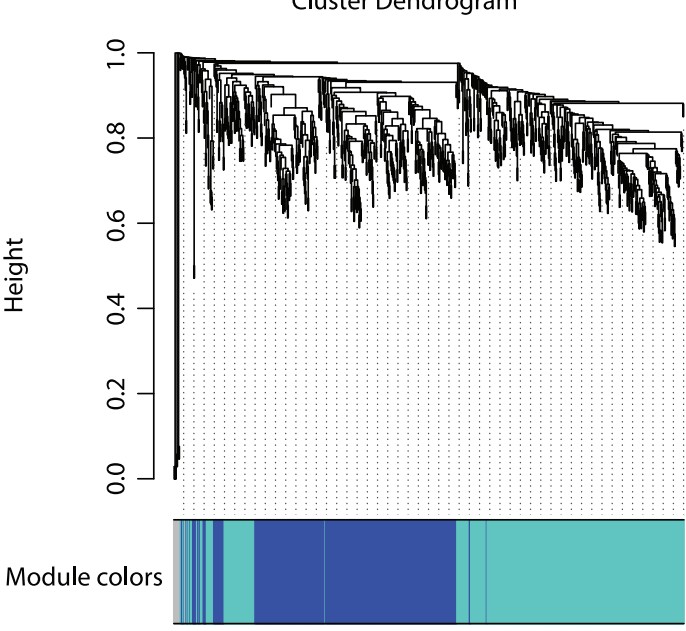

**Fig 4. Cluster dendrogram with dissimilarity based on topological overlap, together with assigned module colors (in this case blue and turquoise) calculated by WGCNA.**

## Module-trait Relationships

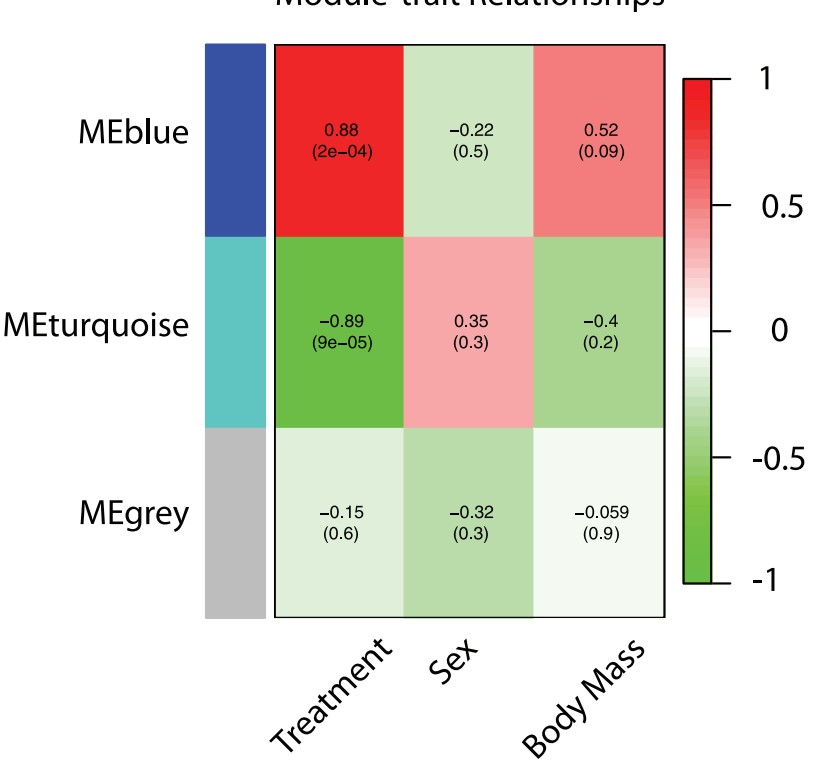

**Fig 5. Module-trait associations with each row corresponding to a module eigengene, each column to a trait.** Each cell contains the corresponding correlation and p-value.

catabolism, which is opposite to the intended goal of tissue repair and rebuilding, however, initial changes such as these are consistent with tissue inflammation, which necessarily precedes tissue repair. In particular, the strong upregulation of four C-C motif chemokines (CCL2, CCL4, CCL8, CCL24), a collection of specialized secondary mediators of inflammation capable of responding to primary inflammatory mediators such as IL-1beta, implies that annuli cells are preloaded and prepared to respond to mechanical loading with classical inflammation mediators.

The distinct fortification of immune and inflammatory genes identified in the blue module by WGCNA is reinforcing evidence that inflammatory mediators are "first responders" to repetitive mechanical loading, whereas the fortification of genes related to skeletogenesis reinforces evidence for recapitulation of skeletal development as is typical of skeletal and connective tissue responses to injury and inflammation.

Although it is unclear exactly how manual therapy might activate annulus cells, the over-representation of membrane and transmembrane transcripts (Tables 3 and 4) is consistent with various models of cellular mechanotransduction that involve outside-in signaling [26] as well as membrane-associated signaling of immune-recognition and inflammation. In particular, there are noteworthy differentially regulated transcripts encoding for integrins and cytoskeleton (e.g., integrin subunit 4), ion pores (e.g., aquaporin 9), and various membrane receptors (e.g., TLR-2) in response to Hyperloading.

Structural molecules notwithstanding, there are signs of anabolic signaling. For example, there is a noteworthy increase in insulin-like growth factor 1 (IGF-1) and a curious concomitant decrease in the IGF-1-antagonist cartilage intermediate layer protein (CILP). Should such

**Table 5. WGCNA module enrichment interfaced with Gene Ontology.**

| Module | Rank | Enrichment P (Fisher Exact) | Bonferoni P | Genes in Term | Term ID | GOntology Term | Term Name |
|---|---|---|---|---|---|---|---|
| blue | 1 | 3.93E-09 | 7.20E-05 | 40 | GO:0019221 | Biological Process | cytokine-mediated signaling pathway |
| blue | 2 | 7.66E-08 | 1.41E-03 | 47 | GO:0071345 | Biological Process | cellular response to cytokine stimulus |
| blue | 3 | 7.81E-08 | 1.43E-03 | 54 | GO:0045321 | Biological Process | leukocyte activation |
| blue | 4 | 9.35E-08 | 1.72E-03 | 64 | GO:0006955 | Biological Process | immune response |
| blue | 5 | 1.08E-07 | 1.98E-03 | 60 | GO:0001775 | Biological Process | cell activation |
| blue | 6 | 9.84E-07 | 1.81E-02 | 40 | GO:0006954 | Biological Process | inflammatory response |
| blue | 7 | 1.02E-06 | 1.86E-02 | 47 | GO:0034097 | Biological Process | response to cytokine |
| blue | 8 | 1.12E-06 | 2.06E-02 | 83 | GO:0002376 | Biological Process | immune system process |
| blue | 9 | 1.19E-06 | 2.18E-02 | 63 | GO:0031982 | Cell Component | vesicle |
| blue | 10 | 2.73E-06 | 5.02E-02 | 53 | GO:0002682 | Biological Process | regulation of immune system process |
| turquoise | 1 | 9.39E-04 | 1.00E+00 | 11 | GO:0048706 | Biological Process | embryonic skeletal system development |
| turquoise | 2 | 1.78E-03 | 1.00E+00 | 10 | GO:0045165 | Biological Process | cell fate commitment |
| turquoise | 3 | 3.07E-03 | 1.00E+00 | 19 | GO:0009100 | Biological Process | glycoprotein metabolic process |
| turquoise | 4 | 3.39E-03 | 1.00E+00 | 9 | GO:0002062 | Biological Process | chondrocyte differentiation |
| turquoise | 5 | 4.27E-03 | 1.00E+00 | 14 | GO:0048705 | Biological Process | skeletal system morphogenesis |
| turquoise | 6 | 4.27E-03 | 1.00E+00 | 14 | GO:0005815 | Cell Component | microtubule organizing center |
| turquoise | 7 | 5.32E-03 | 1.00E+00 | 27 | GO:1901135 | Biological Process | carbohydrate derivative metabolic process |
| turquoise | 8 | 6.41E-03 | 1.00E+00 | 8 | GO:0048704 | Biological Process | embryonic skeletal system morphogenesis |
| turquoise | 9 | 7.19E-03 | 1.00E+00 | 13 | GO:0048839 | Biological Process | inner ear development |
| turquoise | 10 | 7.51E-03 | 1.00E+00 | 19 | GO:0009792 | Biological Process | embryo development ending in birth or egg hatching |

changes result in a net increase in the IGF biosynthesis it could be viewed as a hopeful outcome of manual loading as IGF-1 reportedly has a number of "positive" biological effects, including cell proliferation and matrix synthesis, on intervertebral disc cell metabolism [27]. Even while the overall balance appears to favour catabolism over anabolism, it should be recognized that the targets selectively listed in Table 1 are but a small percentage (20/778 = 2.5%) of the total number of significantly changed transcripts, many of which have unknown functions, hence, it seems fair only to broadly conclude that the cellular response is rich and complex.

MacLean and colleagues used RT-PCT to evaluate a small, select set of mRNA changes in rat (caudal segment) annuli fibrosi (dynamically loaded for 1.5 h) and reported significant and persistent upregulation in the mRNA expression of structural proteins (aggrecan [~ 5-fold], collagen I [~4-fold], and collagen II [~10-fold],) of matrix metalloproteinases (MMP-13

[>10-fold] and MMP-3 [>50-fold]), and the matrix metalloproteinase inhibitor (TIMP-1 [>10-fold]) [20]. Nevertheless, *baseline* (time zero) mRNA expression of all these mRNAs was *not* significantly different compared to endogenous non-loaded controls, but became elevated in all mRNAs except MMP-2 when sampled after 8, 24, or 72 hours after loading [20]. Hence, the methods and system of MacLean et al. were relatively insensitive to changes at baseline, yet clearly defined long-lasting changes in gene expression well after the application of mechanical stimulation. In view of MacLean's findings, the present studies support early initial changes in gene expression that are likely to lead to downstream changes that ultimately could lead to adaptive changes of tissue structure and function.

With respect to discogenic pain, the noteworthy decrease in the expression of galanin receptor 3, which binds the nociception-inhibiting neuropeptide galanin [28, 29] and reportedly has anti-inflammatory activity [30]. These findings provoke speculation that mechanical load might somehow be involved in intervertebral disc neuroinflammation and nociception, which receives some support with the detection of inflammatory pathways in WGCNA analysis.

As with any large dataset, it is difficult to avoid over-interpreting such a long list of differentially regulated transcripts, so it is important to recognize that these studies are a first attempt to define potential targets and pathways that have changed in response IVD loading. Nevertheless, the present studies demonstrate that repetitive manual loading of the living, intact multi-element organ system of the lumbar spine transduces a detectable biological signal, which further reinforces the potential role of mechanobiology in spinal pathobiology, repair, and therapy. Notwithstanding the distinct differential expression of IVD transcripts induced by hyperloading reported here, additional studies are needed to discover whether such changes are robust, repeatable, and lead to functionally significant biological responses to clinical applications of manual therapies.

## Materials and methods

The overall experimental design is outlined in Fig 6. Gene expression of annuli fibrosi exposed to in vivo repeated manual loading (as applied in routine practice) or sham manual loading was evaluated using discovery-based RNA deep sequencing, and a selection of several up-regulated and down-regulated gene targets where chosen for validation by RT-qPCR.

### Animals

Approval for this experiment was provided by the Animal Care and Use Committee at the University of Alberta (573/07/12). Ten domestic Duroc cross (Large White × Landrace) pigs of mixed sex (6 females; 4 males) were included in these studies. Animals ranged in age from 3–4 mos. old and had body mass of 48 kg ± 4 kg (mean ± SD), a size that facilitated the simulation of manual loading in humans. Animals were cared for according to the guidelines of the Canadian Council of Animal Care under the supervision of a veterinarian. Anesthesia was induced using an initial intramuscular injection of azaperone, glycopyrrolate, and a mixture of ketamine- acepromazine-xlyazine, and was maintained for four hours on a mixture of inhaled isoflurane (~1.5%) and oxygen (1%). At the completion of these acute studies animals were euthanized by intravenous injection of barbiturate (Euthanyl, 150 mg/kg).

### Manual spinal therapy

Six animals (four females; two males) were assigned randomly to a "Non-loaded Control" group, and four animals (two females; two males) were assigned to a "Hyperloaded" group. Manual therapy of the spine typical of clinical practice was applied by a trained chiropractor

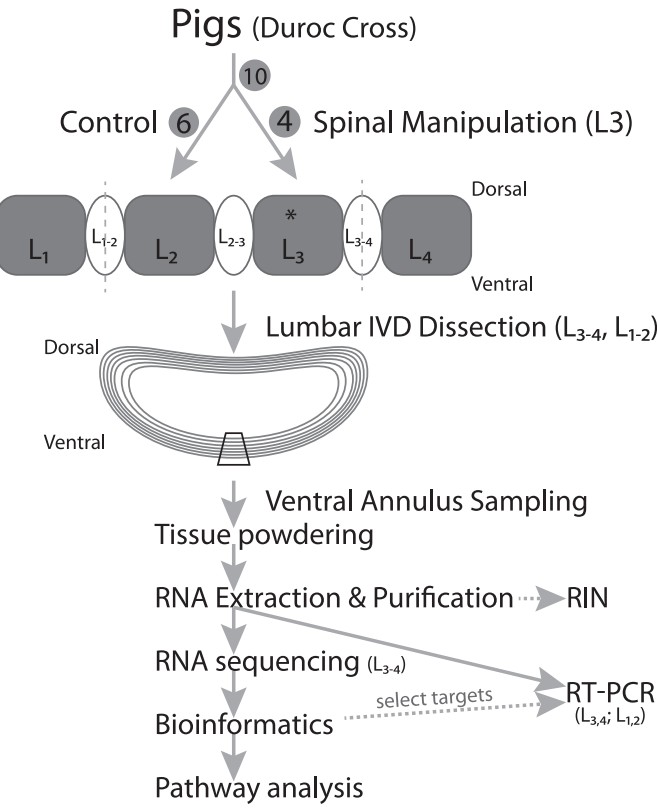

**Fig 6. Flowsheet of overall experimental design.**

(GK) as described previously [31]. The Non-loaded Controls were anesthetized but did not receive any manual loading. Hyperloaded animals received spinal manipulative therapy applied bilaterally to the L3 vertebra once every 1 or 2 minutes for 4 hours; two pigs received loading once a minute (total manipulations = 240), two pigs were loaded once every two minutes, for four hours (total manipulations = 120). The magnitude of manual spinal loading in pigs was calibrated to 400N, the load measured in the human lumbar spine undergoing clinically simulated manual therapy [32]. Briefly, a thin, flexible 10×10 array of 1 cm$^2$ pressure sensors (sensitivity 0.15%, 120 Hz sampling rate [Pressure Profile Systems, Los Angeles]) was inserted between the hand of the manual therapist (GK) and the animal, which monitored real-time loads throughout the experiment. For these experiments, the repetitive frequencies and total number of applications used for these pigs greatly exceed that delivered clinically with the express purpose of inducing an unequivocal positive loading response, hence, we termed this treatment "hyperloading."

## Tissue sampling and processing

At the completion of these acute studies, animals were euthanized and the lumbar spine was dissected to expose the intervertebral discs. The annuli fibrosi from the central ventral (anterior) sector of the L$_{3-4}$ intervertebral discs were dissected and removed (Fig 6). Based on the mid-dorsal application of load, this mid-ventral segment of annulus was chosen for analysis as it was expected to experience the maximum resultant load, thus eliciting the maximal mechanobiological response in the cells of the ventral annuli fibrosi. Annulus segments were cut into small pieces (<30 mg), placed into microfuge tubes, weighed on RNAse-free aluminum foil on

a microbalance, snap-frozen in liquid nitrogen, placed immediately into RNAse-free containers, and stored at -80˚C. Annuli fibrosi was also sampled from the $L_{1-2}$ IVD based on the assumption that this level experienced less load than the $L_{3-4}$ disc at the point of loading. Frozen annuli fibrosi tissue samples were weighed and pulverized into a fine powder using a freezer mill (MikroDismembrator; 2600 rpm for 45 seconds) as described previously [33]. Frozen powdered annulus tissue was immersed immediately in 300 μl RLT buffer containing 1% beta mercaptoethanol for extraction and purification of total RNA using the RNeasy Fibrous Tissue Mini Kit (Qiagen Cat No. 74704). Briefly, RLT-tissue lysate was digested with proteinase K at 55˚C for 10 min, centrifuged 3 min at 10,000 g, and the supernatant transferred and mixed by pipette with a 0.5 volume of absolute ethanol, which was then pipetted over an RNeasy mini-spun column (Qiagen Cat. No. 74104) and centrifuged for 15 sec at >10,000g at 20–25˚C. The column was treated with DNAse I at 22˚C for 15 min, washed twice with RW1 Buffer and centrifuged, then eluted with 25 μL for 5 minutes before centrifugation.

## Evaluating RNA quality

RNA quality was assessed by microspectrophotometry (NanoDrop™ ThermoFisher Scientific) and as RNA Integrity Number (RIN) by microelectrophoresis (Agilent RNA 6000 Nano Kit read by 2100 Bioanalyzer [Agilent Technologies, Santa Clara, USA; No. 5067–1511]). All annulus samples had OD260/280 values exceeding 2.0 and RIN values exceeding 8.0, which indicated high purity of RNA used for sequencing and reverse transcription-quantitative polymerase chain reaction (RT-qPCR).

## RNA sequence analysis and bioinformatics

Twelve samples from annulus level $L_{3-4}$ were used for mRNA (cDNA) sequencing (see below): six samples from six Control animals and six samples from four Hyperloaded animals. The two extra Hyperloaded sample replicates from annulus level $L_{3-4}$ were powdered, isolated, and purified independently to assess repeatability.

## Generation of cDNA libraries and sequencing

The TruSeq RNA sample Prep v2 LS protocol (Illumina, San Diego, CA, U.S.A.) was used for preparation of mRNA libraries. Messenger RNA was purified from total RNA samples using poly-T oligo-attached magnetic beads followed by mRNA fragmentation for first- and second-strand cDNA synthesis. The overhangs, which resulted from the fragmentation of double-stranded (ds) cDNA, were repaired to form blunt ends and a single adenosine was added to the 3′ ends of the blunt fragments to prevent them from ligating to one another during the adapter ligation reaction. Multiple indexing adapters were ligated to the ends of ds cDNA to prepare them for hybridization on a flow cell followed by a PCR amplification step. The libraries were quantified using the qPCR technique and analyzed on a Bioanalyzer 2100 (Agilent Technologies) using a DNA-specific chip. Base calling and demultiplexing of transcriptome sequencing reads were performed using the Consensus Assessment of Sequence and Variance (CASAVA) v 1.6 and Novobarcode software (http://www.novocraft.com/).

## Quality control and alignment of reads

Reads were mapped to the porcine genome Sscrofa11.1 (Ensembl, http://www.ensembl.org/) using JMP Genomics 7.1 (SAS, Cary, NC, USA), allowing two mismatches per read. Total counts and transcripts-per-million (TPM) values were generated for all transcripts. Only

transcripts detected with at least 10 raw reads in all biological replicates for control and/or treated samples were considered present and included in further analyses.

## Data analysis

All data analysis was carried out using TPM values. Quality control, unsupervised clustering, grouped correlation analysis, and Principal Component Analysis (PCA) were performed with JMP Genomics 7.1. Unsupervised cluster analysis is an *assumption-free* approach, hence, each pig was entered individually (i.e., not as treatment and control groups). One-way analysis of variance ANOVA was used to determine differentially expressed genes. Transcripts displaying a fold-change ≥2 and a p-value of ≤ 0.01 were considered differentially expressed. Unsupervised hierarchical cluster analysis was done using the minimum variance method [34], which minimizes within-cluster variance and assigns mutually exclusive subsets of transcriptome profiles from all samples. These analyses independently segregated groups into Non-loaded Controls and Hyperloaded samples (Fig 1). Subsequent pathway analyses were used using these two distinct groups. Principal component analysis (PCA) was performed to distinguish treatment groups.

PANTHER 9.0 (Protein Analysis Through Evolutionary Relationships) Classification System (http://www.pantherdb.org/) and the Database for Annotation, Visualization and Integrated Discovery (DAVID 6.8; https://david.ncifcrf.gov/) [35] were used to classify differentially expressed genes according to their gene ontology (GO) and to statistically determine over- or under-representation of categories (with Bonferroni correction for multiple testing). All remaining (unchanged) transcripts present in Non-loaded Control and the Hyperloaded samples were entered into the analysis as "background." Resulting data were supplemented with additional information from Entrez Gene (www.ncbi.nlm.nih.gov/entrez/query.fcgi?db=gene) and from the literature (http://www.ncbi.nlm.nih.gov/PubMed/). PANTHER was used to identify biological process, molecular function, and cell processes of selected individual genes that were up- and down-regulated in Hyperloaded discs versus Non-loaded Controls.

A systems-level view of genes differentially expressed in Hyperloaded discs was carried out using the WGCNA package in R [36, 37] REF Zhang B and Horvath S (2005); R-project REF). Briefly, pairwise correlations and a power function are calculated to develop a weighted co-expression network of differentially expressed genes, which segregate into discreet clusters termed 'modules.' Pig genes in enriched modules were converted to human orthologs before they could be evaluated (Enrichment P determined by Fisher's Exact test) for functional significance in Gene Ontology (Bioconductor v. 3.12) pathways.

Based on the calculated one-way ANOVA p-value<0.01 for targets with a two-fold change or greater expression (up- and down-regulated targets in Hyperloaded versus Non-loaded Control pigs), a list of potential targets (Table 6) was selected for further validation by RT-qPCR, including: chemokine (C-C motif) ligand 8 (CCL8), chemokine (C-C motif) ligand 2 (CCL2), aquaporin 9 (AQP9), serglycin (SRGN), cartilage intermediate layer protein (CILP), collagen type XI alpha-1 chain (COLL11A1), and fibulin 7 (FBLN7).

## Quantitative RT-PCR analysis of select RNA sequencing targets

Total RNA was transcribed into cDNA using the Stratagene first-strand reverse transcription kit (Stratagene Cat#200420) according to the manufacturer. PCR primers were designed using Primer-BLAST [38] and qPCR amplification of template cDNA was performed in triplicate in a real-time thermocycler (Bio-Rad iCycler) using Sybr Green detection system (iQ™ SYBR® Green Supermix Bio Rad Cat#170–8880). qPCR targets were quantified using the "Fit Point Method" by iCycler Bio-Rad software (2 -ΔΔCT) and normalized to the expression levels of the

**Table 6. PCR primer sequences for select RNA transcripts.**

| Gene | NCBI Reference Sequence | Forward Primer | Reverse Primer |
|---|---|---|---|
| CCL8 | NM_001164515 | GGTGCTTGCTCAGCCAGATT | ACACTGGCTGTTGGTGATTCT |
| CCL2 | NM_214214.1 | CTCCAGTCACCTGCTGCTAT | TGCTGGTGACTCTTCTGTAGC |
| AQP9 | NM_001112684.1 | CAGTCGCGGACATTTTGGAG | AAAGACACGGCTGGGTTGAT |
| SRGN | XM_013990411.2 PREDICTED | CAAGGTTCTCCTGTGCGGAA | AGGGTCAGTCCTTGGAGGTA |
| CILP | NM_001164648.1 | CCCTCTACAAGCACGAGAGC | GGGTTGCAAGGAGCCTATGA |
| COL11A1 | XM_001929372.7 PREDICTED | GGCGATTCTTCAGCAGGCTA | GACCTGGTTCACCACTCTCG |
| FBLN7 | XM_005662277.3 PREDICTED | CCTCCGGATGGCAGAAAGTT | TACCATTGGGAAGACACGCC |

housekeeping gene glyceraldehyde phosphate dehydrogenase (GAP) mRNA. Reaction specificity was ascertained by performing melt-curve analysis at the end of the amplification protocol, and the efficiency of the PCR reaction was evaluated from a dilution series of template (1:1, 1:5, 1:10, and 1:100) using the $R^2$ value of the linear regression of Ct values for GAP, CCL8, and CCL2 (respective $R^2$ values were 0.79, 0.94, and 0.76).

## Supporting information

**S1 Table.**
(PDF)

**S2 Table.**
(PDF)

**S1 Data.**
(XLSX)

## Acknowledgments

The authors are grateful to Xinxin Shao for technical support.

## Author Contributions

**Conceptualization:** John Robert Matyas, Neil A. Duncan, Gregory N. Kawchuk.

**Data curation:** John Robert Matyas.

**Formal analysis:** John Robert Matyas, Claudia Klein, Dragana Ponjevic, Neil A. Duncan, Gregory N. Kawchuk.

**Funding acquisition:** Neil A. Duncan, Gregory N. Kawchuk.

**Investigation:** John Robert Matyas, Claudia Klein, Neil A. Duncan, Gregory N. Kawchuk.

**Methodology:** Claudia Klein, Dragana Ponjevic.

**Project administration:** Gregory N. Kawchuk.

**Supervision:** Neil A. Duncan.

**Validation:** Dragana Ponjevic.

**Writing – original draft:** John Robert Matyas.

**Writing – review & editing:** Claudia Klein, Dragana Ponjevic, Neil A. Duncan, Gregory N. Kawchuk.

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
