## [Decision Letter · Decision Letter 0]

19 Aug 2020

PONE-D-20-22566

Repetitive in vivo manual loading of the spine elicits cellular responses in porcine annuli fibrosi

PLOS ONE

Dear Dr. Matyas,

Thank you for submitting your manuscript to PLOS ONE. After careful consideration, we feel that it has merit but does not fully meet PLOS ONE’s publication criteria as it currently stands. Therefore, we invite you to submit a revised version of the manuscript that addresses the points raised during the review process.

The study is well planned and conducted. The data obtained are interesting, However there are few concern which should be addressed carefully on the specificity of AF using AF markers. In addition the authors has to specifically explain the choice of age groups, method of loading, Number of samples and number of repetitions carefully.  Further interaction of AF with other cells/ factors including from NP will add value owing to the importance of NP cells in IVD well being and regeneration etc. 

We look forward to receiving your revised manuscript.

Kind regards,

Rajakumar Anbazhagan, Ph. D.

Academic Editor

PLOS ONE

Journal Requirements:

2. Please amend your list of authors on the manuscript to ensure that each author is linked to an affiliation. Authors’ affiliations should reflect the institution where the work was done (if authors moved subsequently, you can also list the new affiliation stating “current affiliation:….” as necessary).

3. Please upload a copy of Supplementary Tables S1 and S2 which you refer to in your text on page 6.

Additional Editor Comments (if provided):

The study is well planned and conducted. The data obtained are interesting, However there are few concerns which should be addressed carefully on the specificity of AF using AF markers. In addition the authors has to specifically explain the choice of age groups, method of loading, Number of samples and number of repetitions carefully. Further interaction of AF with other cells/ factors including from NP is very important owing to the importance of NP cells in Bio-mechanical/osmolarity, IVD well being and regeneration etc. Improving the Heat map and adding pathway interaction analysis will add value to the study.

Reviewers' comments:

Reviewer's Responses to Questions

**Comments to the Author**

1. Is the manuscript technically sound, and do the data support the conclusions?

Reviewer #1: No

Reviewer #2: Yes

Reviewer #3: Partly

Reviewer #4: Yes

2. Has the statistical analysis been performed appropriately and rigorously? 

Reviewer #1: Yes

Reviewer #2: Yes

Reviewer #3: Yes

Reviewer #4: Yes

3. Have the authors made all data underlying the findings in their manuscript fully available?

Reviewer #1: Yes

Reviewer #2: Yes

Reviewer #3: No

Reviewer #4: Yes

4. Is the manuscript presented in an intelligible fashion and written in standard English?

Reviewer #1: Yes

Reviewer #2: Yes

Reviewer #3: Yes

Reviewer #4: Yes

5. Review Comments to the Author

Reviewer #1: The manuscript by Matyas et al., describes the effect of repetitive manual loading applied to the lumbar spine on genes expression in annuli fibrosi cells, to determine whether manual therapy has a positive effect on spine. I think that the manuscript is well written and the idea behind the manuscript is good but I have some major concerns regarding this study.

They used 10 animals (6 Non-loaded and 4 Hyperloaded) but then they decided to use 6 controls and 6 hyperloaded sample for the RNA sequencing analysis as the claim in the methods “The two extra Hyperloaded sample replicates from annulus level L3-4 were powdered, isolated, and purified independently to assess repeatability”. Is not clear if this data were included in the analysis at the end.

Moreover they did not assess the health status of the discs in the animals. Why they did not use also degenerated discs or older animals? I think that if the idea is to understand the effect of manual therapy on spine the assumption should be that individuals that are recommended for therapy have back pain and most probably degenerated discs.

Why they analyze the gene expression changes only in annuli fibrosis and did not analyze also nucleus pulposus cells? It is know that NP cells are important for formation and maintenance of IVD and they are considered signaling center in the IVD (Richardson SM et al., 2017; Hiyama et al., 2013; Winkler T et al., 2014), so would have been a more informative study including also information coming from NP cells. They also did not check the purity of preparation (contamination in AF cells from NP or end plate).

Even though the data presented here are indicative of an effect of manual therapy on spine health, additional experiments and data would be needed to support their hypothesis.

Reviewer #2: The manuscript entitled “Repetitive in vivo manual loading of the spine elicits cellular responses in porcine annuli fibrosis” by Matyas et al., analyzed the response of annuli fibrosis for in vivo loading using a transcriptomic approach. The study is interesting and uncovered the differential expression of genes involved in the metabolic process. Although there are certain limitations (due to model and variable response of IVD to loading and corresponding cellular response) in the study which the authors fairly pointed in the discussion studies of similar kind in future will lay the foundation to understand the molecular and biochemical response to physical forces in treating back pain. As the authors rightly concluded there is a need for additional studies in the field to confirm the observed transcriptome regulation to interpret the results clinically.

Overall the manuscript is well written, easy to understand, and adequately detailed for the methods followed. I recommend accepting this manuscript with a minor revision.

Minor suggestions.

1. At the outset how do the authors rule out the observed changes in mRNA regulations are not due to stress applied during the hyper loading procedure?

2. Since the current study can stand as a reference to future studies on IVD loading Is it possible for the authors to group or cluster the differentially regulated genes and show how the regulated genes interact (using tools like DAVID for interaction analysis) with genes differentially regulated in IVD degenerative disease data sets? This will support to claim that the genes regulated after the short term repetitive IVD loading procedure could compensate for the clinical need.

3. The authors can move Figure 1A to supplement and keep Figure 1B as main Figure 1.

4. The authors can represent the qPCR data as a bar graph in Figure 2 along with the RNA seq fold change. Label the genes in the RNA seq Fold change panel. This data can also be combined as panel B and C in Figure 1.

5. I believe Fig 3 is a representation of Table 4, If not what is the difference in the analysis?

6. From the GO analysis, I see the upregulation of genes involved in immune response which is not very well discussed in the manuscript. The upregulated immune genes or pathways could be key for the therapy? please discuss.

7. Also, the pictorial representation of gene ontology in Fig 3 should be verified for the functional terms on Y-axis.

8. The authors cited Table S1 and S2 for differentially regulated genes, but I could not find the supplement file in the submission.

9. Please provide the GEO accession for the data set submission

10. Dataset submitted to OSF needs to be curated carefully for missing gene name and details

Reviewer #3: The Manuscript entitled “Repetitive in vivo manual loading of the spine elicits cellular responses in porcine annuli fibrosi” describes the transcriptomic analysis of ventral annuli fibrosi to single dose of spine manual loading. They use a total of 10 replicates to drive the expression of annuli fibrosi cells. The analysis presented in this paper explains, in brief, the functional roles of transcripts after manual loading. In-depth data analysis, such as pathway analysis would have added more value to the paper. Some issues need to be addressed to improve the manuscript overall,

1. The discussion is quite vague and not comprehensive. A major portion of the discussions are on the limitation of the study. A more detailed description of the biological importance of the differentially expressed genes is required.

2. Can the authors explain in brief the functional role of the Annuli fibrosi cells, why these cells were chosen for the study?

3. How many females and males were used? why 3-4 months old were chosen? How relevant it is to use 3-4 months old pig for this study?

4. ketamine, acepromazine, xlyazine, azaperone, and glycopyrrolate? Were all the drugs combined to anesthetize the pigs? What is the dosage used?

5. To identify all majority of DEGs, at least six biological replicates per condition of experiments is recommended. I was wondering why there is a discrepancy in the number of replicates between control and hyperloaded? The hyperloaded pigs are treated differently and have two subgroups. Did the authors make sure that there are no treatment effects within the subgroups of hyper loaded animals? A PCA plot in this case would be useful. I would suggest including PCA plots to see the differences between the groups.

6. No pathological conditions were induced in the pigs before spinal therapy. Apparently healthy pigs were given spinal therapy and gene expressions were compared. The cellular response to spinal therapy in a pathophysiological condition and a healthy individual may vary. The basis of spinal therapy is to relieve pain and molecular changes seen after spinal therapy may shed light on the molecular mechanism of such therapy. Can the author substantiate the reason to perform in healthy animals?

7. A little more careful explanation of the objective, rationale, and the discussion of the study is justified. The overall objective is not clear. The hyperloading was performed to compare the negative effects of spinal manual therapy or to compare the biological changes to spinal manual therapy in diseased state?

8. I would suggest the authors to perform a pathway crosstalk analysis to further characterize the significantly enriched genes-pathways and also to do a co-expression network analysis to see the interaction signature among the significantly DEG.

9. The heat map is of very poor quality. A brief explanation below each figure would be helpful.

Reviewer #4: The manuscript is about repetitive in vivo manual loads on porcine intervertebral disc using RNA –seq data. This is a very simple but neatly done preliminary study that can be accepted with minor revision.

1. Did authors try to look into in the MRI of the spine of control and hyperloadede animals ? Will they expect any changes ?

2. How many times manual loading was performed ? Can authors explain the significance of timeline of manual loading to sample collection. Where the animals allowed to rest post manual loading ?

3. Did the authors look into aged pigs where the disc degeneration is natural.

4. Figure 1 can be explained in detail in the legends and Figure 2 name the genes analyzed.

6. PLOS authors have the option to publish the peer review history of their article (what does this mean?). If published, this will include your full peer review and any attached files.

Reviewer #1: No

Reviewer #2: No

Reviewer #3: No

Reviewer #4: No

---

## [Author Response · Author response to Decision Letter 0]

20 Jan 2021

All responses to review are uploaded as a file: reviewer comment PLoS1_JRM responses v3

---

## [Decision Letter · Decision Letter 1]

5 Feb 2021

PONE-D-20-22566R1

Repetitive in vivo manual loading of the spine elicits cellular responses in porcine annuli fibrosi

PLOS ONE

Dear Dr. Matyas,

Thank you for submitting your manuscript to PLOS ONE. After careful consideration, we feel that it has merit but does not fully meet PLOS ONE’s publication criteria as it currently stands. Therefore, we invite you to submit a revised version of the manuscript that addresses the points raised during the review process.

Authors should address all comments of all Reviewers. The current revision has "only answers to Reviewer 3"

We look forward to receiving your revised manuscript.

Kind regards,

Academic Editor

PLOS ONE

Additional Editor Comments (if provided):

Author should address all comments of all Reviewers. The current revision has answers to only Reviewer 3.

Reviewers' comments:

Reviewer's Responses to Questions

**Comments to the Author**

1. If the authors have adequately addressed your comments raised in a previous round of review and you feel that this manuscript is now acceptable for publication, you may indicate that here to bypass the “Comments to the Author” section, enter your conflict of interest statement in the “Confidential to Editor” section, and submit your "Accept" recommendation.

Reviewer #4: All comments have been addressed

2. Is the manuscript technically sound, and do the data support the conclusions?

Reviewer #4: Partly

3. Has the statistical analysis been performed appropriately and rigorously? 

Reviewer #4: Yes

4. Have the authors made all data underlying the findings in their manuscript fully available?

Reviewer #4: Yes

5. Is the manuscript presented in an intelligible fashion and written in standard English?

Reviewer #4: Yes

6. Review Comments to the Author

Reviewer #4: (No Response)

7. PLOS authors have the option to publish the peer review history of their article (what does this mean?). If published, this will include your full peer review and any attached files.

Reviewer #4: No

---

## [Author Response · Author response to Decision Letter 1]

12 Feb 2021

The complete set of responses to reviewers is now included in the submission.

---

## [Decision Letter · Decision Letter 2]

22 Feb 2021

Repetitive in vivo manual loading of the spine elicits cellular responses in porcine annuli fibrosi

PONE-D-20-22566R2

Dear Dr. Matyas,

We’re pleased to inform you that your manuscript has been judged scientifically suitable for publication and will be formally accepted for publication once it meets all outstanding technical requirements.

Kind regards,

Rajakumar Anbazhagan, Ph. D.

Academic Editor

PLOS ONE

Additional Editor Comments (optional):

Reviewers' comments:

Reviewer's Responses to Questions

**Comments to the Author**

1. If the authors have adequately addressed your comments raised in a previous round of review and you feel that this manuscript is now acceptable for publication, you may indicate that here to bypass the “Comments to the Author” section, enter your conflict of interest statement in the “Confidential to Editor” section, and submit your "Accept" recommendation.

Reviewer #6: All comments have been addressed

2. Is the manuscript technically sound, and do the data support the conclusions?

Reviewer #6: Yes

3. Has the statistical analysis been performed appropriately and rigorously? 

Reviewer #6: Yes

4. Have the authors made all data underlying the findings in their manuscript fully available?

Reviewer #6: Yes

5. Is the manuscript presented in an intelligible fashion and written in standard English?

Reviewer #6: Yes

6. Review Comments to the Author

Reviewer #6: The authors have addressed my queries and edited the manuscript considering my suggestions. I accept the manuscript in its current format.

7. PLOS authors have the option to publish the peer review history of their article (what does this mean?). If published, this will include your full peer review and any attached files.

Reviewer #6: No

---

## [Editor Report · Acceptance letter]

1 Mar 2021

PONE-D-20-22566R2 

Repetitive in vivo manual loading of the spine elicits cellular responses in porcine annuli fibrosi 

Dear Dr. Matyas:

I'm pleased to inform you that your manuscript has been deemed suitable for publication in PLOS ONE. Congratulations! Your manuscript is now with our production department. 

Kind regards, 

on behalf of

Dr. Rajakumar Anbazhagan 

Academic Editor

PLOS ONE